# Diagnostic and Therapeutic Potential for HNP-1, HBD-1 and HBD-4 in Pregnant Women with COVID-19

**DOI:** 10.3390/ijms23073450

**Published:** 2022-03-22

**Authors:** Mariarita Brancaccio, Cristina Mennitti, Mariella Calvanese, Alessandro Gentile, Roberta Musto, Giulia Gaudiello, Giulia Scamardella, Daniela Terracciano, Giulia Frisso, Raffaela Pero, Laura Sarno, Maurizio Guida, Olga Scudiero

**Affiliations:** 1Department of Molecular Medicine and Medical Biotechnology, University of Naples Federico II, 80131 Naples, Italy; brancacciomariarita2@gmail.com (M.B.); cristinamennitti@libero.it (C.M.); mariellacalvanese99@gmail.com (M.C.); alexgenti98@libero.it (A.G.); gfrisso@unina.it (G.F.); pero@unina.it (R.P.); olga.scudiero@unina.it (O.S.); 2Department of Neurosciences, Reproductive Science and Dentistry, University of Naples Federico II, 80131 Naples, Italy; roberta.musto@outlook.it (R.M.); giuliagaudiello25@gmail.com (G.G.); giulia.scamardella29@gmail.com (G.S.); 3Department of Translational Medical Sciences, University Federico II, 80131 Naples, Italy; daniela.terracciano@unina.it; 4CEINGE, Biotecnologie Avanzate s.c.ar.l., 80131 Naples, Italy; 5Task Force on Microbiome Studies, University of Naples Federico II, 80100 Naples, Italy

**Keywords:** defensins, COVID-19, pregnancy, immune system, interleukins, coagulation, lipoprotein-a, vitamins, biochemistry, laboratory medicine

## Abstract

Pregnancy is characterized by significant immunological changes and a cytokine profile, as well as vitamin deficiencies that can cause problems for the correct development of a fetus. Defensins are small antimicrobial peptides that are part of the innate immune system and are involved in several biological activities. Following that, this study aims to compare the levels of various cytokines and to investigate the role of defensins between pregnant women with confirmed COVID-19 infection and pregnant women without any defined risk factor. TNF-α, TGF-β, IL-2 and IL-10, β-defensins, have been evaluated by gene expression in our population. At the same time, by *ELISA assay* IL-6, IL-8, defensin alpha 1, defensin beta 1 and defensin beta 4 have been measured. The data obtained show that mothers affected by COVID-19 have an increase in pro-inflammatory factors (TNF-α, TGF-β, IL-2, IL-6, IL-8) compared to controls; this increase could generate a sort of “protection of the fetus” from virus attacks. Contemporarily, we have an increase in the anti-inflammatory cytokine IL-10 and an increase in AMPs, which highlights how the mother’s body is responding to the viral attack. These results allow us to hypothesize a mechanism of “trafficking” of antimicrobial peptides from the mother to the fetus that would help the fetus to protect itself from the infection in progress.

## 1. Introduction

In cases of COVID-19, increased levels of inflammatory cytokines and excessive activation of immune cells called “cytokine storms” have been observed [1,2,3]; in particular, cytokine levels were used to assess disease severity, survival, and response to treatment [4,5]. At the same time, antimicrobial peptides (AMPs) are known to be valuable weapons in the body’s defense response against COVID-19 attack [6]. In humans, the most characterized classes of AMPs are the defensins (α- and β-defensins) [7,8], which are among the main players in the innate immunity system, and are involved in the first line of defense, playing a crucial role in infections and inflammations against pathogenic agents (bacteria, viruses, fungi and parasites) [9,10,11,12]. Recently, defensins have been shown to play a leading role in fighting coronaviruses, including SARS-CoV-2 [13]. On this basis, Luan et al. have shown how, for example, human intestinal α-defensin 5 has the ability to inhibit the link between the S1 subunit of SARS-CoV-2 and the ACE2 receptor, binding itself to the latter, forming multiple bonds to hydrogen and blocking viral infection. Furthermore, they also found that high concentrations of human α-defensin 5 (HD5) are present in enterocytes expressing ACE2. This means that these cells have the ability to block the entry of SARS-Cov-2 in the intestine, reducing the incidence of diarrhea in patients with COVID-19 [14]. Moreover, Zhang et al. demonstrated the importance of human β-defensin 2 against respiratory infections, as it is highly expressed in the epithelium that is distributed from the oral cavity to the lungs. This defensin is able to bind RBD, i.e., the binding domain of the SARS-CoV-2 Receptor Binding Domain (RBD), inhibiting its binding with ACE2 [15]. According to these recent results, defensins could represent an arm bell to fight COVID-19 disease.

On the other hand, defensins also play a leading role during pregnancy, which represents a physiological phenomenon unique in nature, consisting of the symbiosis between partially different individuals. In fact, the fetus carries a genetic make-up that is half of paternal derivation [16]. This type of coexistence requires a refined and complex regulation of the immune system, both maternal and fetal, whose purpose is to guarantee an efficient protection against possible infections and to allow the embryonic development process, avoiding the physiological mechanisms of immune reaction which are carried out by the mother’s organism, and which can be harmful to the embryo [17].

In particular, α-defensins 1 and 3 are present in the amnion, chorion, placenta and cervical mucus plug of pregnant women, forming a real physical barrier that has the task of protecting the fetus from possible infections [18]. On the other hand, with regard to β-defensins, it is known that HBD-1 is constitutively expressed in epithelial cells of the genital and respiratory tract [19], while HBD-2 is expressed in the urinary tract, gastrointestinal tract, respiratory system, skin epithelium and has been recently demonstrated in the amniotic fluid of women with pre-term pregnancy [20]. HBD-3 is the major cationic defensin expressed in several epithelia and is also found in saliva and cervico-vaginal fluids [21]. In addition, HBD-4 was found in the stomach, testes and uterus [22]. Finally, HBD-5 and 6 were detected at the epididymal level [23]. Following that, in human physiology we observed a constitutive expression of HBD1 [19], and during activation of the immune system (such as infection, pregnancy, immunity disorders), we can assist to the production of α-defensins 1 [18] and beta-defensins 2, 3, 4, 5 and 6 [20,21,22,23].

In addition, another fundamental aspect during pregnancy is the intake of micronutrients [24]; in fact, it has been observed that vitamins, especially those with anti-oxidative power, can play a key role in the development of the immune system and therefore in the defense of the organism [25,26]. The coronavirus disease-19 (COVID-19) negatively affects the immune system, and in pregnancy is linked with adverse outcomes; these complications may be linked with the infections-mediated deficiency of micronutrients in pregnant women [27].

Following these considerations, this study aims to compare (a) white blood cells and c-reactive protein to evaluate the inflammatory status; (b) coagulation parameters to understand the influence of COVID-19 on the haemostasis balance; (c) Vitamin A and E to verify the oxidative stress; (d) pro-inflammatory cytokine and defensins level to evaluate the different activation of immunity system in pregnant women with confirmed COVID-19 infection and pregnant women without any defined risk factors in order to shed light on the protective role of defensins as potent antimicrobial agents in pregnant women with COVID-19.

## 2. Results

### 2.1. Characteristics of the Study Population

This study involved 22 mothers with COVID-19 and as many controls. The main characteristics of the study population have been described in Table 1. The two groups do not differ in terms of maternal age, pre-partum weight, gestational age and oxygen saturation at admission. Only blood pressure is slightly higher in the mother affected by COVID-19 in comparison to controls, but still falls within the normal range.

### 2.2. Effects of COVID-19 on the White Blood Cells and C-Reactive Protein in Pregnant Women

To evaluate the health status of pregnant women affected by COVID-19 we decided to analyse the white blood cells (Figure 1A–C); in addition, we assessed the levels of c-reactive protein (CRP) (Figure 1D) to understand the severity of the inflammatory state caused by the coronavirus infection. Our results underlined that in pregnant women affected by COVID-19, when compared with healthy pregnant women, there is a significant increase in neutrophils (*p* value < 0.001) (Figure 1A), in monocytes (*p* value <0.01) (Figure 1B) and lymphocytes (*p* value < 0.001) (Figure 1C). At the same time, there is also a significant increase in serum levels of CRP (*p* value < 0.0001).

### 2.3. Evaluation of Coagulation Parameters in Pregnant Women with COVID-19

To understand whether COVID-19 infection in pregnant women may interfere with haemostasis, we assessed the coagulation parameters (Figure 2A–D). Specifically, our evaluations show that there are no significant changes for both prothrombin (PT) and partial thromboplastin (aPTT) (Figure 1A,B). In fact, the same ratio [International Normalized Ratio (INR)] (Figure 2C) which determines an extension of the PT time and the aPTT time does not show any variation. The only significant variation is observed in the levels of fibrinogen; in particular, pregnant women with COVID-19 have a 2.6-fold increase (Figure 1D) compared to controls which have a high significance (*p* value < 0.0001) (Figure 1D).

### 2.4. Dosage of Lipoprotein-a in Serum and Cord Blood of Mothers Affected by COVID-19

To determine if COVID-19 negatively influenced cholesterol transport mechanisms and/or if a cholesterol build-up could be one of the COVID-19 attachment substrates in pregnant women, we measured lipoprotein-a [Lp (a)]. In particular, through the Elisa method, we measured the levels of Lp (a) both in the serum of pregnant women affected by COVID-19 and in their cord blood. The results obtained showed us that the Lp (a) levels are significantly lower both in the serum of women affected by COVID-19 and in the cord blood when compared with the respective controls, as shown in Figure 3. Instead, there is no significance when comparing the serum of women with COVID-19 and the cord blood from them (Figure 3).

### 2.5. Measurements of Vitamin A and E in Pregnant Women with COVID-19

To investigate whether the nutritional status and therefore the serum levels of vitamins could be influenced by COVID-19 and/or manifestation of the infection in pregnant women was caused by an anomaly in the levels of micronutrients, we decided to measure the levels of Vitamin A and E by HPLC in the serum of women with COVID-19. We have focused our attention on Vitamin A and Vitamin E, as they are the main vitamins involved in oxidative and infectious processes. The data in our possession show a significant decrease in serum levels of Vitamin A in women with COVID-19 when compared with controls (*p* < 0.0001) (Figure 4a), while there is no variation in the serum levels of Vitamin E (Figure 4B).

### 2.6. Assessment of COVID-19 on the Gene Expression of Cytokines and Defensins in Pregnant Women

To underline the activation of the immune system of pregnant women affected by COVID-19, we evaluated the levels of gene expression of some cytokines. In particular, we have focused our attention on TNF-α, TGF-β, IL-2 and IL-10. The results obtained show a significant increase (*p* < 0.0001) of all four cytokines in pregnant patients with COVID-19 when compared with controls (Figure 5A–D). As can be seen from the graphs shown in Figure 5, TNF-α increases by 60 times compared to controls (Figure 5A), TGF-β increases 74-fold compared to controls (Figure 5B), IL-2 increases 8-fold (Figure 5C) when compared with the controls, and finally, IL-10 increases 2-fold when compared to the controls (Figure 5D).

At the same time, we evaluated the gene expression of β-defensins (HBD1-4). The results in our possession (see Figure 6A,D), on the one hand, show a significant increase in the levels of gene expression of HBD-1 and 4, while on the other hand, there is no significant variation for HBD-2 and 3 (see Figure 6B,C). Moreover, we can observe an increase in HNP-1 levels (Figure 6E).

Specifically, HBD-1 levels increase by 1.5 in pregnant women with COVID-19 when compared with controls (Figure 6A), while the levels of HBD-4 increase by 6.5 times in pregnant women with COVID-19 if compared to controls (Figure 6D), whereas HNP-1 level increase by 6.5 times in pregnant women with COVID-19 in comparison to controls (Figure 6E).

### 2.7. Estimation of Cytokine and Defensin Levels in Serum and Cord Blood of Mothers Affected by COVID-19

To detect an accumulation of cytokines both in the maternal serum of pregnant women with COVID-19 and in fetuses, we decided to perform an ELISA test for IL-6 and IL-8 on serum and cord blood. As shown in Figure 7, both IL-6 and IL-8 increase in both serum and cord blood taken from patients with COVID-19. In particular, IL-6 in the serum of pregnant women with COVID-19 increases by 2.9 when compared with controls (*p* < 0.0001) (Figure 7A), while IL-6 in COVID cord blood increases 3.6 times when compared with controls (*p* < 0.001) (Figure 7A). There is no significance when comparing pregnant COVID serum and COVID cord blood for IL-6 (Figure 7A). On the other hand, IL-8 increases 2.8-fold in the serum of pregnant women with COVID-19 when compared with controls (*p* < 0.01) (Figure 7B), while IL-8 in COVID cord blood increases 5.3 times when compared with controls (*p* < 0.001) (Figure 7B). Furthermore, if we compare pregnant COVID serum and COVID cord blood with regard to IL-8, a significant increase (*p* < 0.001) is highlighted (Figure 7B).

In addition, to shed light on the role of defensins (HNP-1, HBD-1 and HBD-4) we performed another ELISA assay on the same samples.

In the case of HNP-1, there is a 13-fold increase in the serum of pregnant women with COVID-19 when compared with controls (*p* < 0.0001) (Figure 8A), while HNP-1 in COVID cord blood increases 7.6 times in comparison to controls (*p* < 0.001) (Figure 8A). Moreover, if we compare pregnant COVID serum and COVID cord blood with regard to HNP-1, a slight and significant decrease (*p* < 0.01) is observed (Figure 8A). The same trend was highlighted for HBD-1 (Figure 8B). In fact, HBD-1 increases 18-fold in the serum of pregnant women with COVID-19 when compared with controls (*p* < 0.0001) (Figure 8B), while HBD-1 in COVID cord blood increases 5.3 times in comparison to controls (*p* < 0.001) (Figure 8B). Moreover, if we compare pregnant COVID serum and COVID cord blood with regard to HBD-1, a significant decrease (*p* < 0.001) is observed (Figure 8B). HBD-4, on the other hand, increases 2.4-fold in the serum of pregnant women with COVID-19 when compared to controls (*p* < 0.0001) (see Figure 8C), and at the same time has a 2-fold increase in COVID cord blood when compared with controls (*p* < 0.01) (see Figure 8C). Finally, if we compare pregnant COVID serum and COVID cord blood with regard to HBD-4, a significant increase (*p* < 0.001) is observed (Figure 8B).

### 2.8. Physical Characteristics of the Newborns

The children born to pregnant women with COVID-19 were subjected to a molecular swab which was negative both at birth and at discharge; moreover, in Table 2 we have reported neonatal anthropometric characteristics. In fact, compared to the control population, both the weight and the circumference of the skull are normal. Furthermore, there were no pre-term births (see Table 1).

## 3. Discussion

We report on a prospective cohort of pregnant women affected by COVID-19 within which we were able to perform various observations that improve the understanding of the biology of the disease and the impact on clinical care. The main findings of the present study indicated that pregnant women with COVID-19 had an alert immune system. Indeed, neutrophils, monocytes and lymphocytes were increased; as well as the CRP. These data are in agreement with what is reported in the literature; in fact in the work of Tanacan et al. in women with COVID-19 there is an increase in inflammatory indices [29].

At the same time, no significant changes were observed in haemostasis parameters except for fibrinogen; which is known to increase in pregnancy [30]. In light of this, the increase in figrinogen and its role as a protective factor in our study is supported by the Lp (a) values found in mothers affected by COVID-19. In the case, we highlight a decrease of this apoprotein in the serum of the COVID-19 mothers and in the cord blood of the same in comparison to controls. Since it is well known that higher Lp (A) levels are associated with an increased cardiovascular risk [31], we could suggest that COVID-19 infection does not lead to an increase in such a risk in pregnancy.

Secondly, we measured the Vitamin A and Vitamin E in controls and pregnant women affected by COVID-19, observing a slight decrease in Vitamin A levels in the latter.

Vitamin A has anti-inflammatory activity, contributes to the production, regulation and maturation of immune cells including macrophages, neutrophils, natural killer T cells, dendritic cells, innate lymphoid cells, T cells and B cells. Its deficiency is associated with impaired intestinal immune responses and increases the risk of mortality associated with respiratory infections [32,33]. In addition, during pregnancy, Vitamin A is essential for morphological and functional development and for ocular integrity; moreover, Vitamin A exerts systemic effects on various fetal organs and on the fetal skeleton [34].

Consequently, if, on the one hand, this deficiency could cause a facilitation of engraftment of respiratory tract viruses (e.g., COVID-19), on the other hand it did not compromise the development of the fetus. In fact, the children born showed normal weight, height and head circumference.

Instead, there is no significant change in Vitamin E level, whose data is important, as any deficiencies in Vitamin E denote an increase in radical oxygen species (ROS) and muscle damage. Furthermore, Vitamin E is essential for macrophages and antigen-presenting cells. Moreover, it has been found that Vitamin E deficiencies may represent the substrate for the aggravation of COVID-19 disorders [25,27]. It is likely that the decrease in Vitamin A was offset by the normal level of Vitamin E, and for this reason, the mothers and fetus did not show any problems both during pregnancy and after pregnancy.

However, we cannot exclude that Vitamin E levels could have been influenced by the different intake of multivitamins during pregnancy; on the contrary, Vitamin A is not present is these drugs.

The data obtained from RT-pcr show that on the one hand, the increase in pro-inflammatory factors (TNF-α and IL-2) might explain the increased risk of preterm birth that has been reported in pregnant women affected by COVID-19 infection [29,35], while on the other hand, the increase in IL-10 (anti-inflammatory cytokine) [36,37,38] could represent a maternal response against inflammation, playing a protective role and preventing this outcome.

Moreovere, an increase in TGF-β is observed, which is essential for the transition between the cell-mediated immune response to the humoral one [39]. In addition, there is an increase in HBD1 and HBD4, which is extremely interesting since the first is constitutively expressed in the genital tract [40,41], while the second is inducible in case of infections [22]. Furthermore, we observed an increase in HPN-1, as it points out that there is an ongoing infection. In fact, it has been shown that an increase in HPN-1 coincides with an increase in neutrophils, which supports the presence of an infectious pathology [42].

In light of these results, we can affirm that AMPs are essential in the activation of innate immunity when there is the presence of pathogenic microorganisms [7,8,10,12,43]. So certainly the increase in gene expression of HBD1, HBD4 and HPN-1 in the maternal stage indicates that the maternal immune system is defending itself from the virus’s attack, while at the same time defending the fetus.

Simultaneously, the increase in TNF-α can induce the activation of IL-6, which can perform both pro and anti inflammatory activities [44]. From the data in our possession, we observed an increase in IL-6 both in the serum of the COVID mothers and in the cord blood of the same mothers when compared with the respective controls. This increase highlights the multifunctionality of this interleukin. In fact, it can be secreted by T lymphocytes and macrophages to stimulate the immune response during an infection. The role of IL-6 as an anti-inflammatory cytokine in our study group is given by its effect as an IL-10 stimulating agent [45]. In fact, in our case, in mothers affected by COVID-19 we find an increase in IL-10 since it makes us understand how the maternal immune system is fighting the viral infection in defense of the fetus. Furthermore, the same trend can be observed for IL-8. IL-8 is also known as a chemotactic factor capable of stimulating neutrophils and other granulocytes to migrate to the site of infection [46]. To support this, we have observed an increase in neutrophils, monocytes and lymphocytes in COVID mothers. In addition, the levels of the latter are higher in the cord blood. Recent studies have reported that the fetus’ immune system begins to develop from the second trimester [47,48]. It is likely that the passage from mother to fetus of this chemoctatic factor is further confirmation of how the maternal immune system is in close relationship with the fetus to defend it from virus attacks [49].

Finally, we evaluated the levels of HNP-1, HBD-1 and HBD-4 in both maternal serum and cord blood of COVID-19 and non-pregnant women. The data in our possession show an increase in HNP-1, HBD-1 and HBD-4 in both serum and cord blood of patients with the COVID compared to controls; in addition, the levels of HBD-4, inducible defensin [22], found to be higher in the cord blood taken from pregnant women with COVID-19. This data is extremely interesting, and allows us to hypothesize an AMP trafficking mechanism that would help the fetus protect itself from the infection in progress. In fact, as previously observed over the course of the study, there were no stillbirths or pre-term births and, above all, all the newborns were negative for COVID-19 and did not present any complications caused by the presence of the infection. In our study group, none of these consequences have been verified, and this phenomenon is very likely due to the correct functioning of the maternal immune system, but above all to the trafficking of AMP, which we discovered work as “defenders of the fetus”. Our study, for the first time, sheds light on the role of defensins in pregnant women, in particular regarding the role of HNP-1, HBD-1 and HBD-4 during COVID-19 infection. Further studies conducted on larger sample sizes are recommended to support our findings.

## 4. Materials and Methods

### 4.1. Ethical Approval

The study was conducted according to the ethical guidelines of Helsinki Declaration of the World Medical Association and was approved by the ethics committee (protocol 140/20/ES2COVID19) of the University of Naples Federico II.

### 4.2. Study Design and Study Population

This was a prospective observational case-control study. For this study, we recruited twenty-two pregnant women with COVID-19 and as many controls. All subjects were informed of the purpose and procedures of the study, and written informed consent was obtained from each participant.

Cases were pregnant women with COVID-19 infection admitted to the COVID Unit of Mother and Child Department at University Hospital Federico II between January 2021 and June 2021. Controls were pregnant women with uneventful pregnancies admitted to the Mother and Child Department at University Hospital Federico II during the same period.

COVID-19 infection was documented by the presence of a positive nasopharyngeal swab.

Exclusion criteria were: maternal age <18 years, multiple gestations, pre-existing chronic conditions, gestational diabetes and/or hypertensive disorders of pregnancy, premature rupture of membranes and patients admitted in active labour.

At inclusion, demographic, clinical and anamnestic data were collected on a dedicated dataset.

### 4.3. Samples Collection

We collected a maternal serum sample, a cord blood sample and a placental sample for each included patient.

Maternal blood was collected after inclusion using a BD vacutainer (Becton Dickinson, Oxfordshire, UK) blood collection red tube (with no additives); all patients were asked to respect a 12-h fast before blood collection. After centrifugation, the sample was immediately frozen to −80° C until the time of analysis.

Soon after the delivery and before the afterbirth, an arterious cord blood sample was collected and stored as reported for maternal blood samples.

Placentas were collected immediately after the delivery and sampled in the central area of the placental disc, discarding the maternal decidua and collecting the underlying villi. Samples were stored at −80° C until the time of the analysis.

### 4.4. Biochemical Determinations

White blood cell counts were performed using the Siemens Advia 2120i hematology analyzer (Siemens Healthcare, Munich, Germany) according to the manufacturer’s recommendations. The dosage of c-reactive protein was evaluated on Architect c16000 (c-reactive protein assay, Abbott Diagnostics, Chicago, IL, USA) according to the manufacturer’s recommendations. Coagulation analyses were performed according to the manufacturer’s recommendation using a standard plasma analyzer ACLTOP550^CTS^ (Laboratory Company, Inova Diagnostics, Inc. and Biokit, San Diego, CA, USA). Sera were analyzed for Vitamin A and E concentrations by means of HPLC procedures using the Agilent 1260 Infinity II (ClinRep^®^ Complete Kit “Vitamins A and E” Recipe, Munich, Germany) according to the manufacturer’s recommendations.

All analysis was performed in triplicate in order to guarantee the accuracy of results.

### 4.5. Elisa Assay

Lipoprotein A was measured in the sera by ELISA (Human Lp(a) ELISA (10-1106-01), Mercordia, Uppsala, Sweden) according to the manufacturer’s recommendations. Interleukins 6 and 8, alpha-defensin 1 and beta-defensin 1 were assessed in the sera using ELISA (Human DEFα1 and Human DEFB1 ELISA Kit, Elabscience, Buckingham, UK) according to the manufacturer’s recommendations. Beta-defensin 4 was dosed through ELISA (Human DEFa4, ELK4909 Kit, ELK Biotechnology, Wuhan, China).

All analysis was performed in triplicate in order to guarantee the accuracy of results.

### 4.6. RNA Extraction and cDNA Synthesis

Total RNA was extracted from the placenta (one fragment of 50 mg for each samples) by mechanical homogenisation with Ultra Turrax in Trizol Reagent according to the manufacturer’s protocol (Life Technologies, Carlsbad, CA, USA). The amount of total extracted RNA was estimated measuring the absorbance at 260 nm and the purity by 260/280 and 260/230 nm ratios by Nanodrop (ND-1000 UV-Vis Spectrophotometer, NanoDrop Technologies, Wilmington, DE, USA). For each sample, 1000 ng of total RNA were retro-transcribed with iScriptTM cDNA synthesis kit (Bio-Rad, Hercules, CA, USA), following the manufacturer’s instructions.

### 4.7. Gene Expression by Real-Time qPCR

For real-time qPCR experiments, the data from each cDNA sample were normalised using the human housekeeping gene RLP0 (ribosomal protein lateral stalk subunit P0). The specific primers used for amplification of RLP0, TNF-α, TGF-β, IL-2, IL-10, HBD1, HBD2, HBD3 and HBD4 were designed based on the nucleotide sequences downloaded by NCBI database (see Table 3) using Primer3WEB v.4.0.0. RLP0 primer forward 5′-TGGCAGCATCTACAACCCTG-3′, primer reverse 5′-GACAAGGCCAGGACTCGTTT-3′, HBD1 primer forward 5′-TTTTGTCTGAGATGGCCTA-3′, primer reverse 5′-GGGCAGGCAGAATAGAGACA-3′, HBD2 primer forward 5′-ATCAGCCATCAGGGTCTTGT-3′, primer reverse 5′-GAGACCACAGGTGCCAATTT-3′, HBD3 primer forward 5′-TGAAGCCTAGCAGCTATGAGGATC-3′, primer reverse 5′-CCGCCTCTGACTCTGCAATAA-3′, HBD4 primer forward 5′-AGATCTTCCAGTGAGAAGCGA-3′, primer reverse 5′-GACATTTCTTCCGGCAACGG-3′, TNF-α primer forward 5′-CAAGGACAGCAGAGGACCA-3′, primer reverse 5′-CGTCCCGGATCATGCTTTCA-3′, IL-10 primer forward 5′-TCCATTCCAAGCCTGACCAC-3′, primer reverse 5′-AATCCCTCCGAGACACTGGA-3′ IL-2 primer forward 5′-AACCTCAACTCCTGCCACAA-3′, primer reverse 5′-GCATCCTGGTGAGTTTGGGA-3′, TGF-β primer forward 5′-GGTGAGGAAACAAGCCCAGA-3′, primer reverse 5′- TGCCTCCCAAAAGTGCTAGG-3′; HNP-1 primer forward 5′-CATCCTTGCTGCCATTCTCC-3′, primer reverse 5′-CCTGGTAGATGCAGGTTCCA-3′. Calculations of relative expression levels were performed using the 2^−ΔΔCt^ method [50]; all analyses were performed in triplicate in order to guarantee the accuracy of results.

### 4.8. Data Analysis and Statistics

All statistical analyses were performed using GraphPad Prism 6 (GraphPad Software Inc., La Jolla, CA, USA). Data were expressed as the means ±  standard deviations. The Student’s *t*-test was used to compare the groups, with values of *p* < 0.05 considered significant.

## 5. Conclusions

Pregnancy is a physiological condition characterized by an impaired activation of the immune system. Our results reported that mothers affected by COVID-19 have an increase in pro-inflammatory factors (TNF-α, IL-2, TGF-β, IL-6 and IL-8) and at the same time, in anti-inflammatory agents such as IL-10, when compared to controls. Therefore, we postulate that the increase in pro-inflammatory factors generates a sort of shield to protect the fetus from viral attacks.

Contemporarily, we observed the increase in HNP-1, HBD-1 and HBD-4 both in pregnant women with COVID-19 and in their cord blood, allowing us to hypothesize an interaction between the mother and fetus, which has the ultimate goal of protecting the fetus from infection.

However, this is a preliminary study, and it will in fact be necessary to improve the number of participants and better investigate the molecular mechanisms under defensins’ trafficking.

In conclusion, these results reveal a new scenario from both a diagnostic and therapeutic point of view. In fact, a serum increase in defensins would represent an alarm bell for any infections in progress, but on the other hand, it might be thought that the use of these AMPs in infusion—such as immunoglobulin—during infections with Cytomegalovirus [51,52,53] could be an avant-garde therapy in the case of COVID-19 in pregnancy.

## Figures and Tables

**Figure 1 ijms-23-03450-f001:**
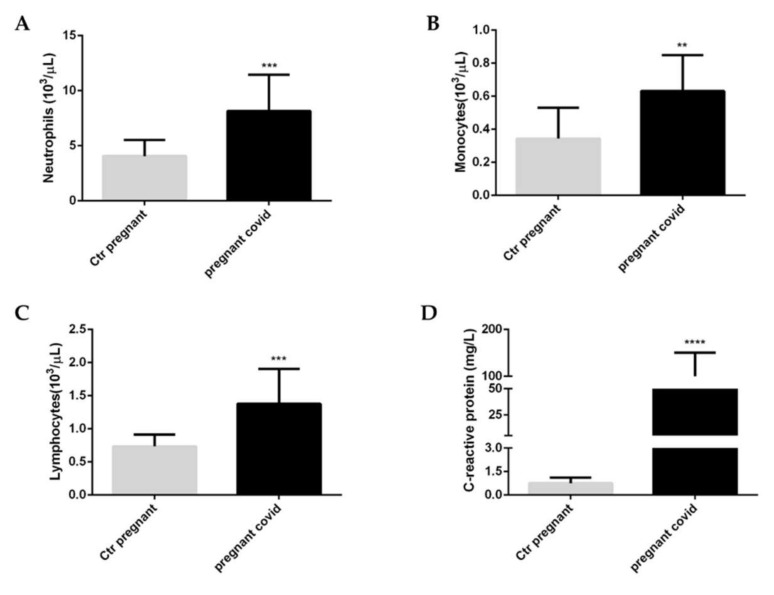
**White blood cells and c-reactive protein in pregnant women with COVID-19**. (**A**) Neutrophils, (**B**) Monocytes, (**C**) Lymphocytes, (**D**) C-reactive protein. The data are expressed as the means ± SDs. The significance was determined by the Student’s *t*-test: ** (*p* < 0.01), *** (*p* < 0.001), and **** (*p* < 0.0001) represent significance compared to Ctr pregnant (pregnant women not affected by COVID-19).

**Figure 2 ijms-23-03450-f002:**
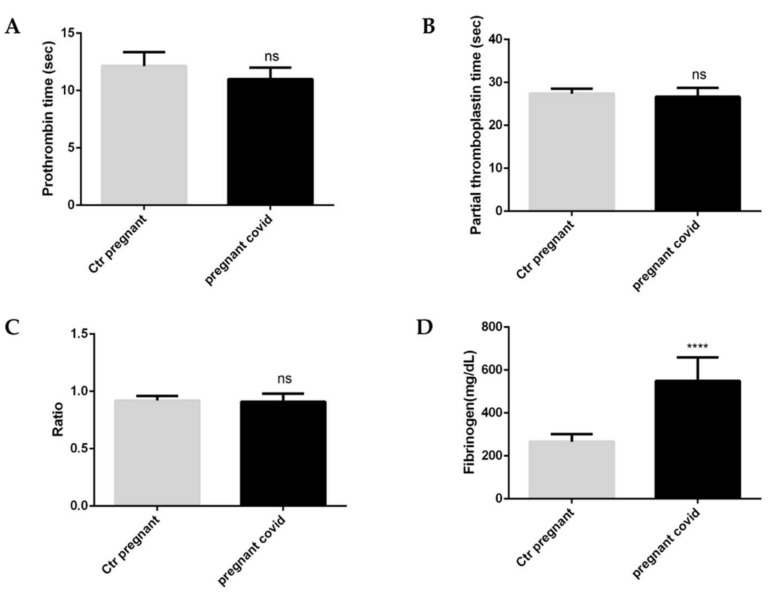
**Coagulation parameters in pregnant women with COVID-19**. (**A**) Prothrombin, (**B**) Partial thromboplastin, (**C**) Ratio, (**D**) Fibrinogen. The data are expressed as the means ± SDs. The significance was determined by the Student’s *t*-test: ns (not significant), **** (*p* < 0.0001) represent significance compared to Ctr pregnant (pregnant women not affected by COVID-19).

**Figure 3 ijms-23-03450-f003:**
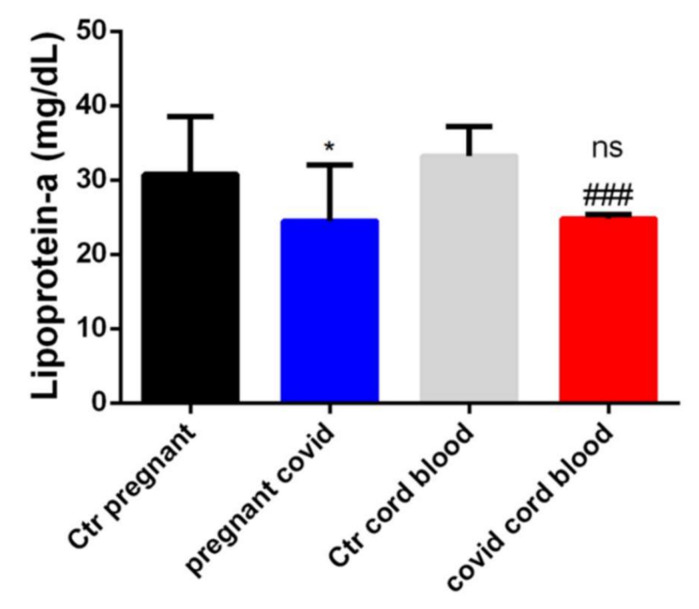
**Evaluation of Lp(a) in serum and cord blood of mothers affected by COVID-19**. The data are expressed as the means ± SDs. The significance was determined by the Student’s *t*-test: * (*p* < 0.05) represent significance compared to Ctr pregnant (pregnant women not affected by COVID-19), ### (*p* < 0.001) represent significance compared to Ctr cord blood (cord blood from pregnant women not affected by COVID-19), ns (not significant) represent significance compared to pregnant COVID (women affected by COVID-19).

**Figure 4 ijms-23-03450-f004:**
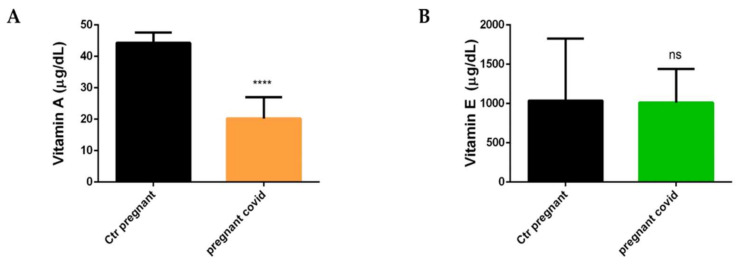
**Dosage of Vitamin A and E**. (**A**) Vitamin A. The data are expressed as the means ± SDs. The significance was determined by the Student’s *t*-test: **** (*p* < 0.0001) represent significance compared to Ctr pregnant (pregnant women not affected by COVID-19). (**B**) Vitamin E. The data are expressed as the means ± SDs. The significance was determined by the Student’s *t*-test: ns (not significant) represent significance compared to Ctr pregnant (pregnant women not affected by COVID-19).

**Figure 5 ijms-23-03450-f005:**
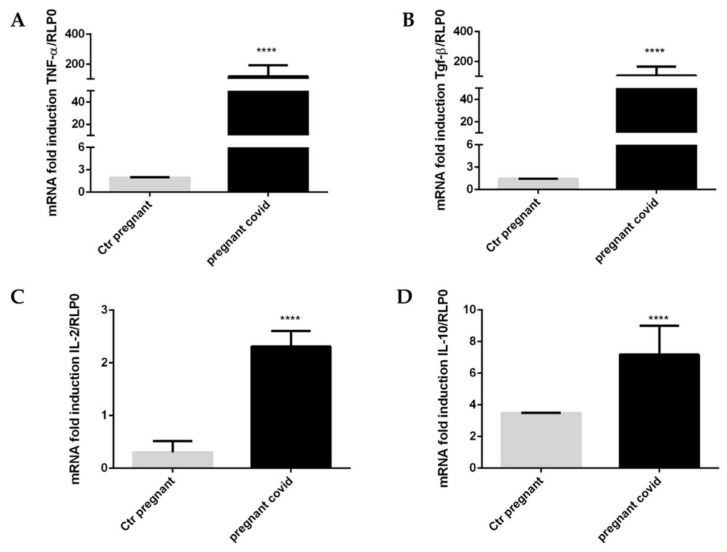
**Effect of COVID-19 on the gene expression of cytokines in pregnant women**. (**A**–**D**) The levels of gene expression of cytokines were assessed by real-time qPCR. The mRNA levels were normalized to RLP0 levels. The data are expressed as the means ± SDs. The significance was determined by Student’s *t*-test: **** (*p* < 0.0001) represent significance compared to Ctr pregnant (pregnant women not affected by COVID-19).

**Figure 6 ijms-23-03450-f006:**
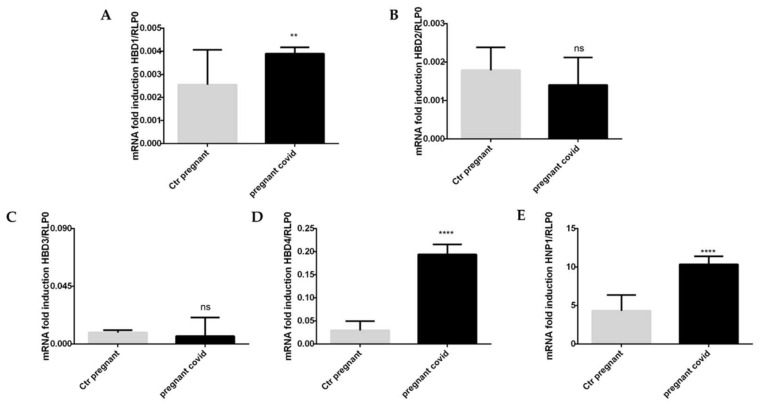
**Defensins gene expression analysis in pregnant women with COVID-19**. (**A**–**D**) The levels of gene expression of β-defensins were assessed by real-time qPCR. The mRNA levels were normalized to RLP0 levels. The data are expressed as the means ± SDs. The significance was determined by the Student’s *t*-test: ** (*p* < 0.01), ns (not significant), **** (*p* < 0.0001) represent significance compared to Ctr pregnant (pregnant women not affected by COVID-19).

**Figure 7 ijms-23-03450-f007:**
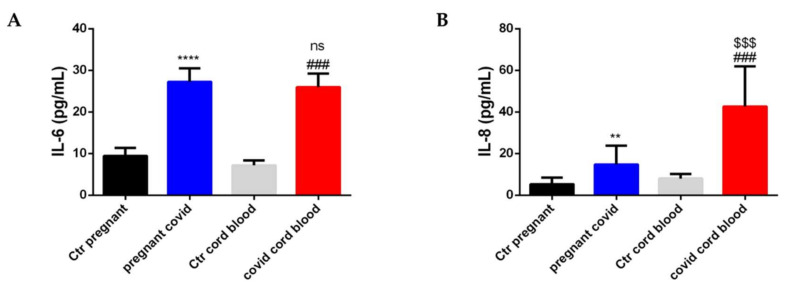
**The level of IL-6 and IL-8 in serum and cord blood of mothers affected by COVID-19**. The accumulation of cytokines was assessed by *ELISA* assay. The data are expressed as the means ± SDs. (**A**) **IL-6** level. The significance was determined by Student’s *t*-test: **** (*p* < 0.0001), represent significance compared to Ctr pregnant (pregnant women not affected by COVID-19); ^###^ (*p* < 0.001) represent significance compared to Ctr cord blood; ns (not significant) represent significance compared to pregnant COVID. (**B**) **IL-8** level. The significance was determined by Student’s *t*-test: ** (*p* < 0.01), represent significance compared to Ctr pregnant (pregnant women not affected by COVID-19); ^###^ (*p* < 0.001) represent significance compared to Ctr cord blood; ^$$$^ (*p* < 0.001) represent significance compared to pregnant COVID.

**Figure 8 ijms-23-03450-f008:**
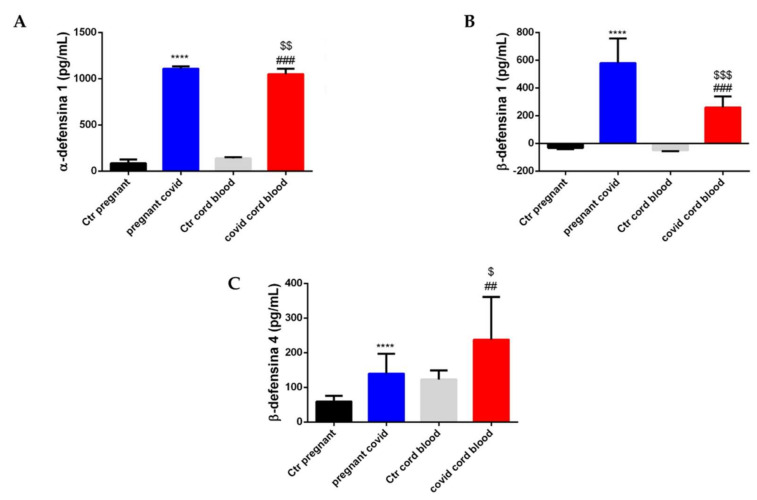
**Accumulation of defensins in in serum and cord blood of mothers affected by COVID-19**. The level of defensins was assessed by *ELISA* assay. The data are expressed as the means ± SDs. (**A**) **HNP-1** level. The significance was determined by Student’s *t*-test: **** (*p* < 0.0001), represent significance compared to Ctr pregnant (pregnant women not affected by COVID-19); ^###^ (*p* < 0.001) represent significance compared to Ctr cord blood; ^$$^ (*p* < 0.01) represent significance compared to pregnant COVID. (**B**) **HBD-1** level**.** The significance was determined by Student’s *t*-test: **** (*p* < 0.0001), represent significance compared to Ctr pregnant (pregnant women not affected by COVID-19); ^###^ (*p* < 0.001) represent significance compared to Ctr cord blood; ^$$$^ (*p* < 0.001) represent significance compared to pregnant COVID. (**C**) **HBD-4** level. The significance was determined by the Student’s *t*-test: **** (*p* < 0.0001), represent significance compared to Ctr pregnant (pregnant women not affected by COVID-19); ^##^ (*p* < 0.01) represent significance compared to Ctr cord blood; ^$^ (*p* < 0.05) represent significance compared to pregnant COVID.

**Table 1 ijms-23-03450-t001:** Characteristics of the study group. Physical characteristics of pregnant women were expressed as mean (±SD).

Characteristics	Pregnant Women with COVID-19	Pregnant Women without COVID-19	*p* Value
Maternal Age	31.5 ± 6 years	34 ± 4 years	0.3348
Pre-partumWweight	67 ± 13 kg	65 ± 11 kg	0.5846
Height	162 ± 6 cm	162.4 ± 5.6 cm	0.8203
Gestational Age	39 ± 1.6 weeks	40 ± 2 weeks	0.0742
Blood Pressure (120–80 mmHg) *	117 ± 6.6 mmHg 77 ± 6.6 mmHg	108 ± 12 mmHg 68 ± 10 mmHg	0.0036 0.0010
Saturation (O_2_)	97 ± 1.5%	97 ± 1.5%	>0.9999

* Reference values of The American College of Obstetricians and Gynecologists (ACOG) [28].

**Table 2 ijms-23-03450-t002:** Characteristics of the newborns. Anthropometric characteristics of newborns are expressed as mean (± SD).

Characteristics	Children Born to COVID-Positive Mothers	Children Born to Non COVID-Positive Mothers	*p* Value
Weight	3.45 ± 0.5 kg	3.45 ± 0.3 kg	>0.9999
Height	50 ± 1.7 cm	50.4 ± 1.3 cm	0.3856
Head Circumferences	35 ± 1.5 cm	35 ± 1 cm	>0.9999

**Table 3 ijms-23-03450-t003:** NCBI accession numbers of the genes analysed.

Gene	Accession Numbers
RLP0	NM_053275.4
TNF-α	NM_000594.4
TGF-β	NM_000660.7
IL-2	NM_000586.4
IL-10	NM_001382624.1
HBD1	NM_005218.4
HBD2	NM_004942.4
HBD3	NM_001081551.4
HBD4	NC_000008.11
HNP1	NM_004084.4

## Data Availability

All the results obtained from the study were reported in the manuscript. There is no additional data.

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
