# Peer review of "Diagnostic and Therapeutic Potential for HNP-1, HBD-1 and HBD-4 in Pregnant Women with COVID-19"

_ijms, 2022, doi:10.3390/ijms23073450_

Round 1
Reviewer 1 Report
The presented manuscript consolidates extremely interesting results regarding the possible involvement of a number of cytokines and antimicrobial peptides from the defensin group as protectors when pregnant women which are infected with the SARS-CoV2 virus. It should be noted that the general direction associated with the global epidemic of this virus does not lose its relevance, in particular, in the aspect of the study of the so-called possible delayed pathologies. In this regard, the results presented can be directly applied in the treatment of this disease in the very near future. A number of questions are outlined below:
- Why did the authors primarily investigate HBDs? Why were theta-defensins not included in the selection?
- Were there exclusively asymptomatic cases among the pregnant women included in the sample? The question is called based on the blood pressure and saturation values shown in Table 1.
- Why was only vitamins A and E quantified in this study? But what about the various vitamins from B groups?
- Please specify which of the HBDs are inducible and which are exclusively constitutive?
- Characteristics of newborns, presented in table 2, are almost identical in terms of average values between both groups. How can this be explained, given the rather large variation in maternal scores?
- Which HBDs have shown antiviral activity? From the point of view of the authors, how can the activation of the biosynthesis of a whole group of defensins occur in response to infection with a virus? And could this be more of a secondary infection? Like bacterial?
Author Response
Dear Editor,
thank you for the Reviewer’s Report about our manuscript entitled “Diagnostic and therapeutic potential for HNP-1, HBD-1 and HBD-4 in pregnant women with Covid-19”, submitted to International Journal of Molecular Sciences. We have appreciated the comments received by the Reviewer and yourself and have carefully re-considered them in preparing a new version of the manuscript.
A point-by-point response to the comments is attached below.
We believe that the manuscript is now significantly improved thanks to the Reviewer’s inputs.
We hope that the new version of the paper deserve publication on International Journal of Molecular Sciences.
Best regards,
Prof. Dr. Maurizio Guida and Dr. Laura Sarno
Point-by-point response.
Reviewer 1
The presented manuscript consolidates extremely interesting results regarding the possible involvement of a number of cytokines and antimicrobial peptides from the defensin group as protectors when pregnant women which are infected with the SARS-CoV2 virus. It should be noted that the general direction associated with the global epidemic of this virus does not lose its relevance, in particular, in the aspect of the study of the so-called possible delayed pathologies. In this regard, the results presented can be directly applied in the treatment of this disease in the very near future. A number of questions are outlined below:
Response:
First of all, we thank Reviewer 1 for comments and suggestions that improved the quality of our paper.
- Why did the authors primarily investigate HBDs? Why were theta-defensins not included in the selection?
We clarify why we primarily investigate HBDs in the introduction. Moreover, theta defensins are found in non-human primates such as gorillas and / or chimpanzees and therefore we have not been considered. (Conibear AC, Craik DJ. The chemistry and biology of theta defensins. Angew Chem Int Ed Engl. 2014; 53 (40): 10612-10623. Doi: 10.1002 / anie.201402167; Tongaonkar P, Tran P, Roberts K, Schaal J, Osapay G, Tran D, Ouellette AJ, Selsted ME. Rhesus macaque θ-defensin isoforms: expression, antimicrobial activities, and demonstration of a prominent role in neutrophil granule microbicidal activities. J Leukoc Biol. 2011 Feb; 89 (2): 283-90. Doi: 10.1189 / jlb.0910535. Epub 2010 Nov 17. PMID: 21084627; PMCID: PMC3024902)
- Were there exclusively asymptomatic cases among the pregnant women included in the sample? The question is called based on the blood pressure and saturation values shown in Table 1.
The hospitalized mothers were all symptomatic; the pressure, albeit in the normal range and tending to high if we compare it with healthy subjects not affected by Covid-19. The pressure is not out of the normal range as no mother had previous cardiovascular disorders and / or diseases such as obesity and diabetes which can often affect the increase in blood pressure in pregnancy.
In the materials and methods session, lines 393-397 we specified the inclusion criteria for the study.
- Why was only vitamins A and E quantified in this study? But what about the various vitamins from B groups?
During the pregnancy, normally, among the B vitamins, B6 plays an important role in the development of the central nervous system of the fetus; moreover vitamin B6 can alleviate pregnancy nausea. This vitamin, together with folate, is evaluated by gynecologists in particular in the first three months of pregnancy. In case of deficiency supplementation is prescribed. In our study, the mothers were close to giving birth, and affected by covid-19; so we decided to dose two antioxidant vitamins that are important to the immune system. Furthermore, vitamin A in pregnancy cannot be supplemented due to its teratogenicity (Guillonneau M, Jacqz-Aigrain E. Les effets tératogènes de la vitamins A et de ses dérivés [Teratogenic effects of vitamin A and its derivates]. Arch Pediatr. 1997 Sep; 4 (9): 867-74. French. Doi: 10.1016 / s0929-693x (97) 88158-4. PMID: 9345570); therefore, this monitoring seemed to be of interest to us, precisely to understand how their possible deficiency and / or increase could be used by the maternal immune system.
- Please specify which of the HBDs are inducible and which are exclusively constitutive?
Accordingly your consideration, we have specified the function of HBDs in the Introduction session, lines 89-92.
- Characteristics of newborns, presented in table 2, are almost identical in terms of average values between both groups. How can this be explained, given the rather large variation in maternal scores?
This study aimed to shed light on how a maternal immune system, not compromised by previous pathologies and / or not influenced by pathologies strictly related to pregnancy such as (gestational diabetes, preeclampsia, etc.) can defend the fetus from possible viral attacks such as 'Covid-19 infection.
Our hypothesis of trafficking of AMPs between the mother and the fetus supports the data reported in table 2; in fact, all newborns to Covid-19 positive mothers had similar characteristics to those born to healthy mothers.
Therefore, as can be seen from the table, the two populations were homogeneous.
This data therefore suggests that an external supplementation of defensins, in the event of a viral attack, could help the mother to carry the pregnancy to term and above all to have a correct development of the fetus.
- Which HBDs have shown antiviral activity? From the point of view of the authors, how can the activation of the biosynthesis of a whole group of defensins occur in response to infection with a virus? And could this be more of a secondary infection? Like bacterial?
Antimicrobial peptides are an abundant and diverse group of immune response active molecules that form the first line of defense against pathogens. Over the years, it has been seen that these AMPs can be activated even in conditions that put the immune system under severe stress (such as pregnancy which is considered allogeneic transplant-like; in which the maternal immune system must adapt to the fetus where 50% of the genetic heritage is of paternal derivation).
In our study, mothers with no prior clinical disease and/or relapsing bacterial infections; they were hospitalized because they were symptomatic of Covid-19 and in the vicinity of childbirth.
During viral infections such as covid-19 (Kudryashova, E., Zani, A., Vilmen, G., Sharma, A., Lu, W., Yount, JS, Kudryashov, DS Inhibition of SARS-CoV-2 Infection by Human Defensin HNP1 and Retrocyclin RC-101. J. Mol. Biol. 2021, 167225; Luan, J., Ren, Y., Gao, S., Zhang, L. High level of defensin alpha 5 in intestine may explain the low incidence of diarrhoea in COVID-19 patients. Eur. J. Gastroenterol. Hepatol. 2020; Zhang, L., Ghosh, SK, Basavarajappa, SC, Muller-Greven, J., Penfield, J., Breweer, A., Ramakrishnan, P., Buck, M., Weinberg, A. Molecular dynamics simulations and functional studies reveal that hBD-2 binds SARS-CoV-2 spike RBD and blocks viral entry into ACE2 expressing cells. BioRxiv 2021), the defensins being part of the first lines of defense activated and their expression in the circle increases.
Therefore, from the data in our possession we have shown how covid-19 infection puts the maternal immune system under "strain", which puts in place a remarkable series of defenses to protect the fetus.
The mothers in the study, in fact, did not have abortions, and they are all full-term pregnancies in which the fetus has developed correctly.
All modification was highlight in yellow in the text.
Reviewer 2
Thank you for giving the opportunity to review this manuscript "Diagnostic and therapeutic potential for HNP-1, HBD-1 and HBD-4 in pregnant women with Covid-19". I appreciate the authors effort for drafting a very well written manuscript and my decision is to accept the manuscript after incorporating some changes.
Response:
First of all, we thank Reviewer 2 for comments and suggestions that improved the quality of our paper.
- The abstract section is too long. My suggestion is to shorten it and include only main findings.
Accordingly your consideration, we have reduced the abstract section
- The introduction section can be improved after clubing the short paras.
Thank you, in order to satisfy your suggestion we have improved the introduction
- Please revise the conclusion section by adding study limitations.
Accordingly your consideration, we have modified the conclusion
All modification was highlight in yellow in the text.

Reviewer 2 Report
Thank you for giving the opportunity to review this manuscript "Diagnostic and therapeutic potential for HNP-1, HBD-1 and HBD-4 in pregnant women with Covid-19". I appreciate the authors effort for drafting a very well written manuscript and my decision is to accept the manuscript after incorporating some changes.
- The abstract section is too long. My suggestion is to shorten it and include only main findings.
- The introduction section can be improved after clubing the short paras.
- Please revise the conclusion section by adding study limitations.
Author Response

(The authors gave the same response as above.)

Round 2
Reviewer 2 Report
Accept.